# The Paradox in the Ecological Side of Corporate Entrepreneurship Sustainability: A Research Agenda and Policy Direction

Yangfan Lu [1], Abdella Kosa Chebo [2,3,*], Shepherd Dhliwayo [3] and Semu Bacha Negasa [4]

1 School of Public Administration, South China University of Technology, Guangzhou 510640, China
2 Faculty of Business and Economics, Kotebe University of Education, Addis Ababa 31248, Ethiopia
3 School of Management, University of Johannesburg, Johannesburg 524, South Africa
4 Department of Management, Harambe University, Adama 2932, Ethiopia
* Correspondence: abdela.kossa@kmu.edu.et or akosa@uj.ac.za; Tel.: +251-912-444616

**Abstract:** The ecological devastation observed in the 21st century requires everyone's participation, including corporates. Many companies have, therefore, incorporated these ecological concerns into their sustainability decisions. We reviewed studies on the nexus of ecological and entrepreneurial sustainability in the context of corporates (large enterprises). The review focuses on clarifying concepts and building a conceptual framework to enhance a better and comprehensive understanding of the ecological side of corporate entrepreneurship sustainability. Through a systematic review of 53 selected papers, we provide inputs for integrating the ecological aspects into company policies and philosophy, in order to build a green business that balances business opportunities with environmental commitments. The study initiates new research agendas by creating a new construct—sustainable corporate ecological entrepreneurship (SCEE).

**Keywords:** ecological; entrepreneurship; sustainability; corporate

## 1. Introduction

An enterprise's environmental concerns require that it makes appropriate adjustments to its environmental practices. In recent studies, researchers have combined environmental concepts with entrepreneurship, and theoretical and contextual links have been partially identified [1]. Some of the previous research on this topic focused on a wide range of issues, such as environmental entrepreneurship (e.g., [2]); environmental impacts (e.g., [3]); environmental economics (e.g., [4]; sustainable green entrepreneurship (e.g., [1]); and corporate entrepreneurship (e.g., [5]. However, only a few of these relate to a corporate culture (philosophy) that embodies commitment to environmental sustainability. Conceptually, the notion of ecological entrepreneurship allows the exploitation of business opportunities while respecting and fulfilling environmental obligations [6–9]. It focuses on combining profit orientation with the creation of a greener business world.

The 21st century has witnessed numerous environmental problems [9], some of which are attributed to unsustainable business practices [10]. Since industrial processes are causes of climate change, business-related activities are one way to address these challenges [11]. That is, to achieve greener industrial growth [12,13], new production models that minimize negative externalities [14] and consider the ecological sector as a strategic path [10,15] are required. As a result, entrepreneurs have started to integrate and consolidate their environmental concerns and set up eco-friendly businesses [16]. Companies are increasingly considered one of the key contributors to successfully addressing environmental threats [3,17]. They do so by institutionalizing the environment in their processes and systems [8] with the primary aim of creating a clean environment ([1]). Therefore,

understanding the relationship between entrepreneurship and the environment is very important [18] and requires further investigation.

Nevertheless, the main motive that drives companies to be sustainable is the desire to make a profit [19] while largely avoiding environmental problems [20]. In addition, industrialization, considered important for economic development but potentially harmful to the environment [3], creates a contradiction between economic development and environmental protection [15]. Conversely, scholars from different disciplines agree that business is not only a main culprit of the aforementioned problems, but also plays a crucial role in overcoming them [21]. Therefore, one of the main concerns of the authors of the current paper is the link between entrepreneurship and the environment [1], because the question of how entrepreneurship can contribute to solving environmental problems [3,21] is receiving increasing attention. Moreover, organizations such as NGOs [11] and individuals have recognized the need to improve the ecological environment [22]; they tend to put significant pressure on companies [23] to address problems by inventing new sustainable practices [24]. This is supported by improvements in internal company structures and resources that can affect the level of aspiration for environmental sustainability [8].

The theory of sustainable ecological entrepreneurship is relatively new and leaves many avenues open for further research and study. However, various theories have partially taken this aspect into account. Among these, stakeholder theory and the natural-resource-based view (NRBV) can be used to theoretically justify the relationship between green business direction and business performance ([8]). According to stakeholder theory [25], companies must strive to meet the goals of all stakeholders and thereby promote social and institutional sustainability. To some extent, both theories take into account the social importance of the ecological environment. Related to this is neoliberalism, expounded by Milton Friedman, which states that maximizing economic profit should be the primary goal and that corporate social responsibility (CSR) is thought to hurt financial profitability. Later, neo-Malthusian environmentalists argued that, in previous decades, economic growth through entrepreneurship had not been matched with ecosystem preservation [9]. This indicates the presence of contradictions, and the topic of entrepreneurship has received relatively little attention in ecologically oriented studies [18].

In particular, the targeted consideration of ecological concerns in business decisions only receives limited attention, while the idea of sustainability is broader and includes social, economic, and ecological aspects. In addition, most previous theories focused on separate studies on corporate entrepreneurship or green entrepreneurship. Given these, researchers have accepted the correlation between business and the environment in the field of entrepreneurship [1]. However, the concept of sustainable corporate ecological entrepreneurship (SCEE) on the ecological side of corporate entrepreneurship is less discussed and not well treated, which requires further investigation.

By incorporating updated results from the recent literature, this review provides some scientific contributions to the topic of SCEE. First, although some literature reviews have previously been conducted on the subject [1–20], they differ from the current one in terms of the research purpose, the concepts of the construct, the type of verifications, and the methods used. There is also interest in learning about the latest advances in the subject.

Second, a small pilot survey does not provide comprehensive studies related to CSEE because they are multidisciplinary and scattered across different study areas. The study of corporate entrepreneurship mainly focuses on financial returns and competitive advantages [26], while there is a lack of literature connecting corporate entrepreneurship with ecological entrepreneurship and sustainability at the same time. Sustainable entrepreneurship relies on the assumption of a connected scenario of individual, organizational, and contextual factors [27]. However, our review only focuses on the factors related to corporate organizations. In addition, most of the theories and normative frameworks proposed so far come from established fields such as social entrepreneurship and environmental economics [28]. The study of sustainable entrepreneurship is vague, as it covers wider issues such as societal, environmental, and economic aspects (the triple bottom line). Al-

though this is paramount in empirical studies, it is crucial to focus on one specific aspect to carry out a detailed analysis of the specific topic. Hence, this study is unique in that it focuses on one of the elements of the triple bottom line (TBL)—environmental/ecological entrepreneurship. That is, this review builds an integrated and comprehensive framework for corporate sustainable environmental entrepreneurship by systematically structuring the fragmented concepts related to SCEE, in order to provide valuable insights and a future research agenda.

Third, considering previous empirical studies, which have largely examined large companies, there are still underdeveloped studies that specifically focus on the environmental business practices of SMEs [13]. Although these studies focused on either large or small companies, we reviewed studies conducted in both large and small companies at the same time. This review is also not limited in terms of countries or regions but considers the literature reflecting the practices of corporate entrepreneurship related to ecological sustainability in different parts of the world. It also compares and specifies the practices of developed and developing countries. In addition, this review reveals the characteristics and different conditions under which companies integrate the elements of sustainable ecological entrepreneurship. Therefore, the comprehensive framework of corporate ecological entrepreneurship sustainability will provide new insights and stimulate debate among policy makers and future researchers.

Given the above facts, the purpose of this study is to provide an overview of the developments and inconsistencies in the literature on SCEE, and to provide a theoretical overview of the concept of sustainability in relation to corporate entrepreneurship and ecological entrepreneurship. In line with this purpose, this study tends to answer research questions such as: (1) What are the key dimensions explored in previous research related to SCEE? (2) Is there an opportunity to consider the specific construct of ecological sustainability in corporate-level entrepreneurial decisions? (3) What are the theoretical foundations underlying the field of SCEE? (4) Is there sufficient background that helps to construct the conceptual framework? (5) What new research agendas exist around CEE?

## 2. Theories and Literature

### 2.1. Theoretical Elucidation

Still, it is difficult to form comprehensive theoretical foundations on the subject of SCEE. Regarding the role of entrepreneurship, previous studies offer various theories, including Frank Knight's risk-bearing theory, Alfred Marshall's theory of entrepreneurship, and Schumpeter's theory of entrepreneurship. Milton Friedman's approach later refers to a neoliberal system and a conventional mode of production that are characterized by profit and competition. Theories such as resource-based views and dynamic capabilities also explain the strategic integration of sustainability into the entrepreneurial thinking of organizations. Although these theories provide a basis for corporate sustainability at the company level, they are not extended to the environmental aspects of companies in their strategic decisions. The NRBV relates the organization's competitive advantage to the natural environment [27]. Against this background, in 1999, Hart and Milstein emphasized the potential of the interplay between entrepreneurship and sustainable development, which is gradually evolving into a broader approach of the triple-bottom-line perspective [20]. These theories are considered broad aspects of economic, environmental, and societal sustainability. Therefore, there is still a requirement for a theory that specifically addresses the ecological aspects in corporate-level entrepreneurial and strategic decisions.

Proponents of the sociological approach examine how the origins of environmental economics and the principles of ecology relate to entrepreneurship and the business spirit, which address the role of ecopreneurs in society and the way that ecopreneurship can be used going forwards as a vehicle to change social structures [11]. Under the dynamic capabilities approach, environmental proactivity is considered a dynamic capability [27]. Moreover, the theory of planned behavior (TPB) is also used to study the green behavior of individuals in different domains [29]. In comparison, these theories have taken environ-

mental aspects into account, although some of them relate the environment to individual entrepreneurial actions, while others are linked to organizational environments.

The institutional perspective provides reasons why governments encourage all members of society to actively support sustainability initiatives such as green entrepreneurship [30]. The stakeholder theory states that firm or individual activities are either influenced by the firm or affect the way the firm operates [8]. However, both theories have not sufficiently emphasized the ecological aspects of corporate entrepreneurship. Starting from an integrated approach, a combination of entrepreneurship, management, and neo-institutional theories is proposed to construct a theory of sustainable development [27]. Later, the theory of ecological modernization (EMT) emphasizes that entrepreneurs are the key agents of change in the transformation process to prevent an ecological crisis [31]. Conversely, Pacheco et al. [32] see sustainability as a green prison for entrepreneurs [27]. Given the above facts, theories such as EMT, NRBV, TPB, and institutional theories are used as the basis for this study.

*2.2. Brief Characteristics of Previous Literature*

In the last decade, numerous academic papers have examined different aspects of sustainable entrepreneurship, including corporate sustainable entrepreneurship [23], sustainable ecological entrepreneurship [30], sustainable entrepreneurial opportunities, and the context and success factors for sustainable entrepreneurship [22], etc. In the literature, entrepreneurial endeavors within an existing organization are often referred to as "corporate entrepreneurship" (CE) [33]. CE is treated as a behavioral construct independent of entrepreneurial orientation (EO) [34]. On the other hand, sustainable entrepreneurship focuses on the conservation of nature, life support, and community, in the pursuit of perceived opportunities [28]. Then, the essence of sustainable entrepreneurship becomes the discovery of new opportunities while committing to social and environmental responsibility [35]. This leads to entrepreneurship being recognized as a solution rather than a cause of environmental degradation [28]. In this respect, it becomes important to create an innovation-friendly environment and encourage entrepreneurial behavior in existing companies, since this consolidates the ecological environment towards sustainability.

Faced with unsustainable economic development models that have had adverse environmental impacts, and despite the promotion of short-term economic prosperity in recent decades, people have come to realize that substantial changes are needed to improve the ecological environment [22]. Based on these demands, entrepreneurs have begun to give greater importance to environmental issues [36]. However, despite the growing interest in ecopreneurship, the academic literature on the subject is still in its infancy [3]. Potluri and Phani [37] believe that environmental concerns can offer entrepreneurs a win–win situation in terms of energy-saving, material reuse, and lower recycling costs [1]. Gevrenova [38] also reported evidence of the essential role that green companies play in the pursuit of greenness and sustainability, contrary to the notion that companies are unimportant to environmental threat [17]. There is, therefore, a need to go beyond the study of sustainable entrepreneurship, which focuses on the integration of ecological aspects, to stimulate further discussions.

Conceptually, in the relationship between the third pillar of TBL, the environment, and corporate entrepreneurship, the literature has used different terms to indicate similar concepts such as environmental entrepreneurship, ecopreneurship, and green entrepreneurship [23]. Nonetheless, issues related to green entrepreneurship are an emerging field in entrepreneurship studies, and various definitions of green entrepreneurship have been provided [26]. The clearest distinction has been made between eco-entrepreneurship (ecopreneurship), social entrepreneurship, institutional entrepreneurship, and sustainable entrepreneurship [28]. For example, Kraus et al. [39] defined ecological entrepreneurship as the process of identifying, analysing, and seizing opportunities to minimize a company's exploitation of the natural environment and create benefits for future societal and economic

needs. However, there is still a need to clarify the SCEE concepts and explore the ecological side of sustainable entrepreneurship in corporate firms.

*2.3. Towards Conceptual Framework Development*

The growing importance of environmental issues and sustainability is helping to shape new trends at the enterprise level [35]. Several studies have ana-lysed sustainable entrepreneurship from the perspective of its drivers and motivations. However, there is a need to define the determining factors and outcomes related to the integrated concept of CEE sustainability. Ecological entrepreneurship involves a green outcome or output [18]. Ecological challenges require a better knowledge of both the drivers and the consequences of ecological entrepreneurial activity [30]; for example, entrepreneurs should be aware of the impact their businesses have, directly or indirectly, on the environment [20]. From an ecological point of view, innovation management no longer only involves the coordination between individual factors, but also offers direct benefits to companies, both internal (e.g., the improved ethical behavior of employees) and external (e.g., positive public image) [13]. Internal drivers such as reducing costs, the ecological footprint, and environmental risk are taken into account, while external pressures such as the public, customers, competitors, and a positive corporate image are also considered [23].

Some studies try to categorize the green management motivations of firms, including identifying competitiveness, legitimacy, and environmental responsibility as the main motivations for green management [23]. It is concluded that government support can be provided in a tangible form (e.g., granting subsidies) or through intangible mechanisms (e.g., activating interactions between environmental entrepreneurs and other key actors) [1]. Studies have linked business motivations such as cost reduction and operational efficiency to the procurement and adoption of greener, higher performing industry solutions [40]. Innovation is also essential to enable a sustainable recycling process [41].

In addition to motivation, there is also pressure that compels companies to adopt eco-logical entrepreneurship. Historically, environmental issues have typically been ad-dressed through legislation, legitimizing, ethical, or competitive initiatives and incentives [27]. As a catalyst for green entrepreneurship, government commitments to supportive and respon-sible institutional policies have the potential to catalyze change and encourage greater investment in innovative and responsible practices [40]. Other studies have identified how the environmental pressures driving the adoption of sustainability practices are exerted by stakeholders in the supply chain, competitors, organizations in the region, and the public administration, and how the strongest pressures come from the regulatory environment and organizations [23]. As a process, the recent sub-stream of corporate entrepreneurship research focuses on the involvement of external partners, as well as significant corporate resources in innovation-generation processes through the use of new business models [42]. The ecological sector is also considered a strategic means to adapt to change [10], including ecological sustainability.

Among the outcomes of companies' ecological entrepreneurial processes, sustain-able entrepreneurship at the macro level strengthens the connection between economy, society, and environmental values [26]. Similarly, ecological entrepreneurship can boost economic activity, increase productivity, maximize competitiveness, and create cutting-edge jobs [40]. At the enterprise level, seeking collective benefit, maintaining communities, and contributing to network development are considered inputs to enterprise performance [43].

## 3. Methodology

Both quantitative and qualitative research approaches through a semi-systematic re-view were used to conduct this study and provide comprehensive knowledge on the topic of SCEE. This is because a systematic review only allows for the inclusion of empirical studies, resulting in important conceptual and theoretical work missing. In addition, a systematic review imposes strict requirements on the search strategy and limits the inclusion of studies with different concepts from different research groups and different disciplines [44].

Therefore, the semi-systematic review is selected, as it provides transparency through the synthesis of the relevant studies by overcoming the problems mentioned that are associated with the full systematic review.

### 3.1. Study Framework and Search Scope

Before starting the review, we checked whether there were similar works on the subject of SCEE to avoid double reviews. Accordingly, we mainly extracted the literature from Google Scholar (GS), since the topic is multidisciplinary. In addition, we checked both forwards and backwards citations to confirm whether all articles were fetched or not. In addition, we followed a four-step methodology recommended by Denyer and Tranfield [45] to ensure study transparency, consistency, and accuracy. These methods include developing review questions and setting the conceptual boundary; demonstrating the search boundary by establishing a review scope; performing the identification, screening, and selection processes using the PRISMA flow chart; and applying the synthesis and analysis.

Afterwards, the review questions were developed. When determining the scope of the review and the database for searching documents, GS was selected because the coverage of WoS and Scopus in the social sciences and humanities is not good [46], while GS is advantageous because it searches all citations from multiple sources and does not differentiate between subject areas [47]. Then, the inclusion and exclusion criteria were used, and the keywords used for search purposes were linked using the Boolean logic approach. Regarding the exclusion criteria, we only considered articles published internationally in English. Concerning the inclusion criteria, relevant works that were missed were included using the snowball and bibliographic methods. Accordingly, the reviewers listed the relevant articles from the bibliography in each article they reviewed for inclusion. From this, the conceptual boundary was established based on the integration and conceptual linking of these concepts.

### 3.2. Search Strategy

The advanced search option was used to perform a keyword search. This keyword search was conducted by two independent researchers. Since the focus was on corporate firms, the search was limited to corporate enterprises' perspectives. Accordingly, the thematic areas employed included 'sustainability', 'corporate entrepreneurship, and 'ecological entrepreneurship'. The specific terms used to search included ("corporate entrepreneurship") AND ("ecological entrepreneurship" OR "eco-entrepreneurship" OR "green entrepreneurship" OR "environmental entrepreneurship) AND ("sustainability" OR "sustainable" OR "development" OR "long term growth" OR "continuity). A combined or joint search was then carried out between the selected subject areas to ensure the development of comprehensive knowledge on the subject.

### 3.3. Study Selection and Eligibility Criteria

The inclusion criteria for selecting articles were that only peer-reviewed articles in English were included; and documents from other sources such as books, book chapters, conference papers, technical reports, and other non-peer-reviewed publications were removed. That is, locally published articles and articles published in languages other than English language peer-reviewed articles were not considered. Two independent reviewers were then involved in collecting the data based on the eligibility criteria. When there were disagreements between the two data collectors, the differences were resolved through discussion. Accordingly, the independent reviewers screened and selected all articles that met the inclusion criteria. Later, the duplicate articles were removed by manually checking them. All the steps that were followed are shown in the PRISMA flow chart in Figure 1. Gray literature was used in the search process to add missing literature the literature from GS. The identification, selection process, and eligibility are summarized in Figure 1.

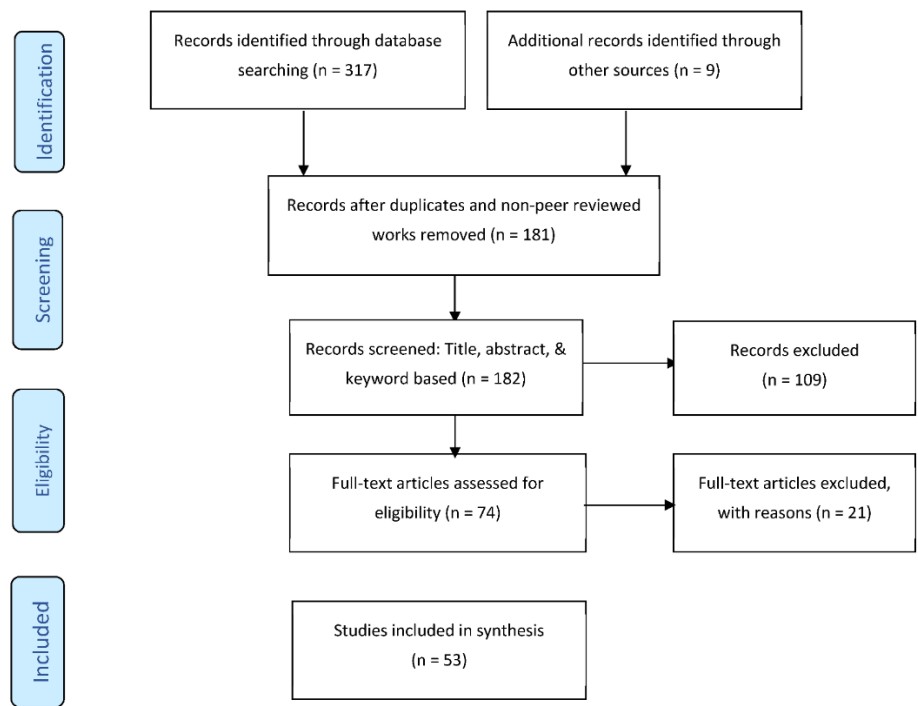

**Figure 1.** Selection process and eligibility.

Regarding the steps followed in the selection criteria, first, the duplicates and works other than the peer-reviewed journal articles were removed, and 182 articles were remnants among the total of 326 articles extracted. The quality of the extracted articles was checked by focusing on articles published in reputable journals, using only peer-reviewed articles, and avoiding duplicates. That is, items with different meanings were eliminated by the selection criteria. Some of the articles contained a different meaning, and others were not related to the research question and were deemed irrelevant to the current study; therefore, they were eliminated. Later, articles were evaluated based on their title, abstract, and keywords, and papers unrelated to the purpose and subject matter of this study were excluded. Accordingly, a total of 109 articles were removed, and 74 articles qualified for further evaluation. We then assessed the full paper, leaving 53 articles for final analysis.

*3.4. Data Extraction and Quality Assessment*

To ensure the quality of the research, the researcher documented the literature findings, the selection of keywords, and the evaluation of the results. A quality-related concern was reduced by considering only peer-reviewed articles and the journals that publish their articles in the publicly available electronic databases. After this, the data were extracted by two independent reviewers. The two researchers then independently searched using the same keywords and found the same results, ensuring the robustness of the review processes. However, a disagreement between the two reviewers on the use of terms was discussed with the third reviewer, and a consensus was reached. To ensure quality following a systematic approach, both authors differentiated the articles according to their relevance, scoring 0 for articles with no relevance to objectives, 1 for articles with little relevance, 2 for articles with fundamental relevance, and 3 for those with deep relevance [48]. To maintain quality, only items rated 2 and 3 were included.

**4. Results and Discussions**

Different insights from the reviewed literature were organized using quantitative de-scription and narrative synthesis. Using a quantitative approach, issues such as publication trends per year, areas of study, journals, approaches used, and geographic areas of publication were summarized. The narrative approach involves organizing the results

into different themes to develop an overall picture of the essential knowledge of the field of study and provide frameworks for further analysis. Therefore, the data collected from different kinds of literature were analyzed both qualitatively and quantitatively.

### 4.1. Descriptive Results

We performed an analysis based on the year of publication, publishing journals, study area, the field of study, and employed methodologies.

Recently, there has been a significant development regarding the publication of studies on SCEE (Figure 2). This indicates that the area is a new and developing area that requires further study. From the search, only one article was found before 2008. Therefore, 2008 was taken as a starting year.

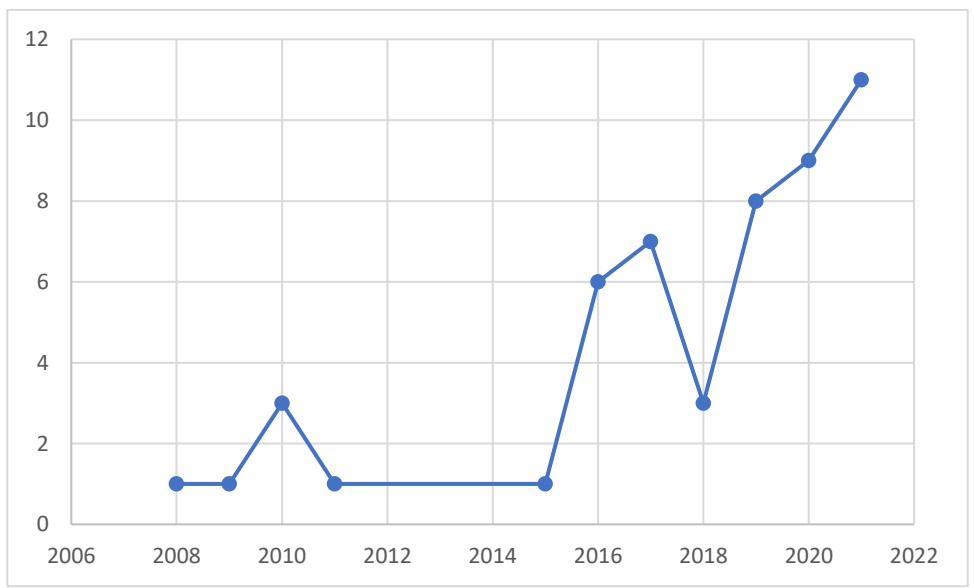

**Figure 2.** Year of publications.

Sustainability is the leading journal in publishing the study of SCEE (N = 8), followed by the Journal of Cleaner Production (5) and Ecological Economics (N = 3) (Table 1). Regarding the country-based analysis, most of the studies in the area of sustainability of corporate eco-logical entrepreneurship were carried out across countries (N = 16). In particular, most of the cross-country studies are conducted in developed countries in Europe. Using Global Entrepreneurship Monitor data collected from 53 countries, Horisch et al. [21] indicate that environmental orientation is widely employed as a source for assuring the legitimacy of entrepreneurial endeavors. Additionally, they demonstrated that only OECD countries' environmental taxes have a discernible impact on the environmental orientation of entrepreneurial endeavors. Over the past half-century, environmental protection issues have become development priorities in developed countries [26]. For instance, the study carried out in Poland by Chwikowska-Kubala et al. [49] did not find that economic activities affected environmental practices. However, there is a lot of attention in this area from developing countries such as China.

Aside from the developing countries, researchers and academics from developing countries have started to address the issues of corporate sustainability and ecological entrepreneurship. For instance, based on the data from 235 new Chinese green firms, the empirical results suggest that green entrepreneurial activities enable them to acquire a green performance advantage over their competitors [6]. However, there are only a few studies in regions such as Africa, which makes it necessary to conduct studies focusing on countries in Africa, due to weak reasoning and poor strategies in practice. This is because developing countries have been challenged by a shortage of skilled workers and a shortage of university graduates in general, especially those educated in science,

technology, engineering, and mathematics (STEM) skills [15]. Notably, although scholars in the field have provided evidence linking ecological entrepreneurship and sustainability in developed countries, a lack of evidence and academic emphasis in developing countries such as Saudi Arabia raises questions about the effectiveness and transferability of such development proposals [40].

**Table 1.** Descriptive results.

| Top Ten Publishing Journals | | Area of Studies | | Field of Studies | |
|---|---|---|---|---|---|
| **Journals** | **No. of Articles** | **Countries** | **No. of Articles** | **Disciplines** | **No. of Articles** |
| Sustainability | 8 | Cross-country | 16 | Business and Economics | 35 |
| Journal of Cleaner Production | 5 | China | 5 | Multidisciplinary | 13 |
| Ecological economics | 3 | USA | 4 | Environmental and Natural Science | 3 |
| Business Strategy & the Environment | 2 | Poland | 3 | Social Sciences | 2 |
| Entrepreneurship & Sustainability Issues | 2 | South Africa | 2 | Technology and Engineering | 1 |
| Int. J. Environ. Res. Public Health | 2 | Saudi Arabia | 2 | | |
| Journal of Business Ethics | 2 | Nigeria | 2 | | |
| Journal of Business Venturing | 2 | Iran | 2 | | |
| Small Business Economic | 2 | | | | |
| World Review of Entr., Man. and Sust. Devel. | 2 | | | | |

Although the study is multidimensional, the majority of research related to corporate ecological entrepreneurship sustainability is conducted in business economics fields (N = 35), followed by multidisciplinary studies (N = 13). This showed a good step in linking business-related areas and activities with environmental sustainability. Many studies consider the field of environmental science, although they are integrated into other fields and included as multidisciplinary studies.

The result shows that most research papers use an empirical approach (N = 39) and the rest are theoretical or conceptual (N = 14) (Table 2). This indicates that the subject still needs to be explored by considering a variety of cases to further develop the discipline and clarify the theoretical aspects. In addition, further empirical studies are needed to determine the antecedents and consequences of sustainable corporate ecological entrepreneurship (SCEE).

**Table 2.** Type of research and designs employed.

| | Research Design | | | |
|---|---|---|---|---|
| **Research Type** | Explanatory | Descriptive | Exploratory | Total |
| **Empirical** | 23 | 5 | 11 | 39 |
| **Conceptual/Theoretical** | 1 | 11 | 2 | 14 |
| **Total** | 24 | 16 | 13 | 53 |

*4.2. Major Themes/Dimensions*

Several dimensions were identified by the authors who conducted work on SCEE. Most of these dimensions relate to either sustainable entrepreneurship or environmental aspects. In particular, most of the dimensions identified can be categorized under corporate



entrepreneurship or ecological concerns, or both terms together. Regarding this subject matter, green entrepreneurship, including green innovation and sustainable entrepreneurship, is a well-researched area in this field. More specifically, the themes identified include business environment turbulence [50]; green entrepreneurship [1,3,8,10,15,40,51]; green innovation [1,41]; green economy [31]; green business [13]; green product innovation [52]; sustainable entrepreneurship [1,23,24]; SEO ([27]); the nexus between sustainable entrepreneurship and pollution [53]; and sustainable supply chain management [52]. Almost all of the above concepts are related to either entrepreneurial or environmental sustainability. This indicates the necessity of linking entrepreneurial activities with environmental factors to achieve the dual objectives of business and the environment. Thus, these important concepts have to be integrated and empirically tested to clearly identify the nature of relationships.

Some authors deal with business issues and environmental concepts integrally. These authors connect entrepreneurship with various aspects of the ecological environment, such as environmental passion, pollution, and green environments. Nevertheless, there is a need to establish a clear link between business concerns and environmental aspects to ensure the sustainability of the company and the ecological environment at the same time. For example, the triple bottom line takes into account more comprehensive aspects of social, ecological, and economic aspects, which prevents specific consideration of ecological aspects in entrepreneurial decisions at the company level. Since these main issues are considered simultaneously, the amount of consideration of environmental issues within companies will be minimal. In addition, the broad concept of SEO, which has been studied by many researchers, has taken organizational rather than environmental aspects into account. Therefore, we argue that companies have to develop a specific program and practice that takes environmental sustainability into account when addressing corporate strategies related to corporate sustainability. This study reiterates the need to continue working on the new concepts of sustainable entrepreneurial ecological entrepreneurship as a focus topic.

Previously, various researchers have defined terms related to the sustainability of corporate ecological entrepreneurship. Some of the most widely used terms are green entrepreneurship, sustainable entrepreneurship, and environmental entrepreneurship. For example, Gevrenova [38] and Maziriri et al. [31] define green entrepreneurship as an economic activity whose products, services, production, or organizational methods have a positive impact on the environment. Green entrepreneurs need to prioritize social responsibility and environmental issues during the process of developing conceptual products, technologies, and services. In general, green entrepreneurship is characterized by ecological dependence on green consumers and on political support. However, green entrepreneurship is mostly used as a comprehensive concept that combines ecological entrepreneurship and sustainable development [15]. Sustainable entrepreneurship is related to an investigation into how opportunities to create future goods and services are discovered, created, and exploited, by whom and with what, and their economic, social, and ecological consequences. Moreover, Pacheco et al. [32] viewed sustainable entrepreneurship as the discovery, creation, evaluation, and exploitation of opportunities to create future goods and services that are consistent with sustainable development. This shows that the integrated construct of SCEE is highly linked to the exploitation of environmental opportunities through conserving the environment in return. Therefore, businesses benefit from conserving the ecological environment, which is believed to ensure sustainable institutional development.

Furthermore, Piwowar-Sulej et al. [3] define environmental entrepreneurship (also known as ecopreneurship) as the process of entrepreneurship applied to companies that solve environmental problems or operate sustainably. However, issues related to green entrepreneurship remain an emerging area in entrepreneurship studies, and various definitions have been provided [26]. We then argued that all of the above definitions would not suffice for a new construct of SCEE. For example, the concept of corporate decision making

is not addressed in all of the above definitions. Some of the definitions limit business decisions to goods and services, while others take into account the broad concept of sustainability, which encompasses social, environmental, and ecological aspects at the same time. Accordingly, we have operationalized a new construct called SCEE as a corporate-level program that incorporates ecological aspects into strategic business decisions. That is, it includes a process and practices for incorporating specific ecological dynamics into the determination of corporate strategies at the enterprise level, with respect to institutional and environmental sustainability.

### 4.3. Antecedents and Consequences of SCEE

Thematically, this study mainly focuses on green entrepreneurship practices in corporates and on the sustainability of entrepreneurship. Although studies such as those by Iqbal et al. [53] have explored the relationship between sustainable entrepreneurship and the ecological environment, there is still a need to examine the causes and consequences of integrating the two concepts. That is, the determining factors and consequences must be reviewed. Factors impacting the entrepreneurial actions of companies concerning the ecological environment were identified and categorized. Accordingly, previous studies such as Alwakid et al. [30] have linked the concept of green entrepreneurship to factors such as environmental action, environmental awareness, and time orientation. In general, we have categorized the antecedents into pull and push factors. For instance, determinants such as economic and business opportunities [1,12,20]; ethical motives [13]; green entrepreneurial spirit [10,40]; innovative technologies [40]; innovative goals and commitment [19]; cost reduction and the efficient use of resources [54]; government subsidies [55]; and benefits to other people [23,36,43]) can be considered as pull factors (Figure 3). They are the main factors that lead firms to practice sustainable ecological entrepreneurship.

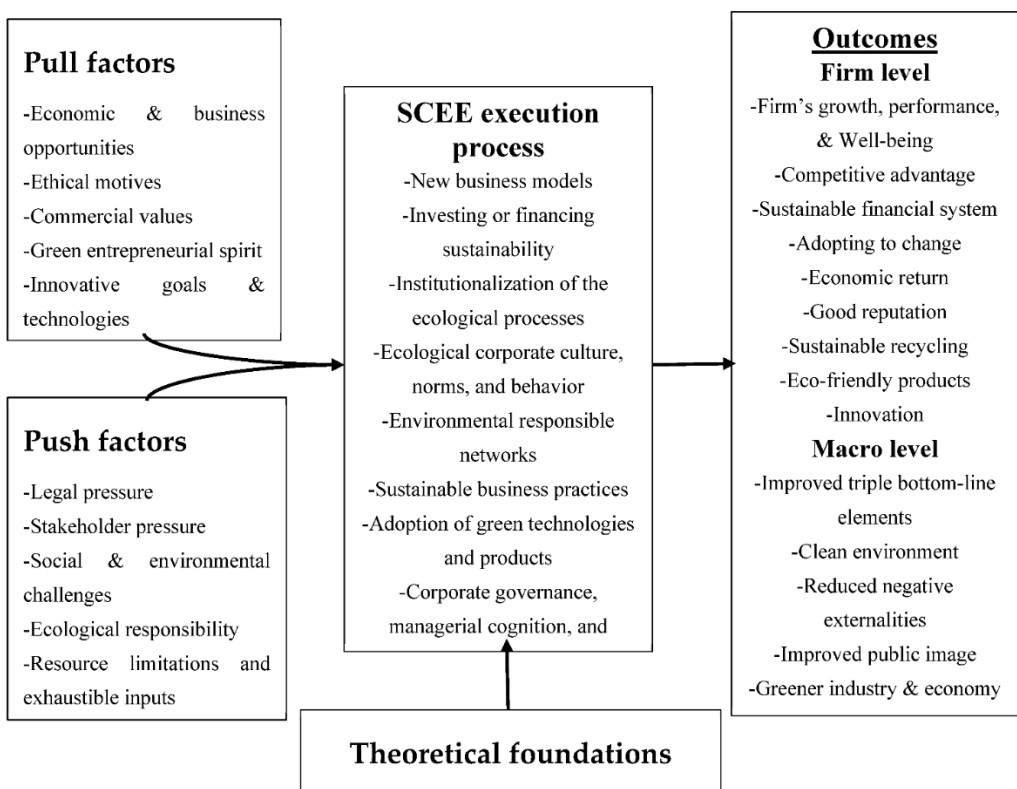

**Figure 3.** Antecedents and consequences of SCEE.

In summary, there are different motives beyond the adoption of SCEE. As one of the company's goals, the economic opportunity that arises from the proper use of the ecological

environment appeals to companies to adopt ecological corporate sustainability practices. There are also various business opportunities related to environmental protection that attract companies. This could be related to the commercial values generated by practicing green entrepreneurship and sustainability issues. Beyond commercial and economic values, companies benefit from minimizing costs through the efficient use of scarce resources, which requires innovative goals and innovative technologies. In addition, companies must be ethical and motivated to help others, leading them to produce and use environmentally friendly products. Businesses are also concerned with the proper use of resources to minimize production costs and improve efficiency. On the other hand, some companies promote green entrepreneurship because of government subsidies. Above all, the presence of a green entrepreneurial spirit among company members and social engagement are highly encouraged for companies.

Aside from the motives that encourage corporate enterprises towards the adoption of ecological and entrepreneurial sustainability, there are push factors that enforce them towards the adoption of such practices; these are legal pressure and regulations [6,11,13,35,40,56]; stakeholder pressure [13,23]; business competitive pressure [23]; social and environmental challenges [11]; ecological responsibility [35]; resource limitations and exhaustible inputs [40]; and negative corporate externalities [57] (Figure 3). To make it clear, the strict rules and regulations imposed by the government and other stakeholders push companies to adopt sustainable ecological business practices. Therefore, companies tend to address stakeholder responses to the environmental aspects of business practices. This is due to strong pressure from the government to control pollution and emissions, as well as the pressure from competitors and green consumers. By forming a dialogue with stakeholders (e.g., government agencies, consumers, suppliers, and retailers), and by following competitors' environmental strategies, this can be addressed. Eco-consumers, for example, initiate corporations' efforts at ecological corporate sustainability because they prefer environmentally friendly products. There are also challenges from society and the environment itself. For example, the risk of ecological degradation and the depletion of natural resources is becoming a concern for the companies that push them to engage in ecological entrepreneurship and sustainability by manufacturing eco-friendly products. In addition, there are pressures from negative corporate externalities, such as emissions and pollution, as well as exhaustible resources.

Influenced by the above factors, the practice of corporate ecological entrepreneurship ensures sustainability and develops positive externalities through developing and extracting new sustainable business models [2,9,42]; investing in sustainability [12,13,39]; institutionalizing the environmental processes [8]; knowledge transfer and integration [7], using digital technologies [2]; developing ecological corporate culture, norms, and behavior [38]; resource commitment [58]; and networks for environmental problems [10]. It is also ensured through corporate governance [19]; managerial cognition [55]; and sustainable business practices [36]. More specifically, firms are developing and practicing a new business model that integrates the two concepts to overcome environmental concerns and achieve business goals. By implementing a new business model, management ensures that the environmental process is being institutionalized; environmental corporate culture and behaviors have been developed; and environmental concerns are integrated into corporate governance, management understanding, and corporate philosophy. This is done by integrating corporate resources such as information, knowledge, and technology into the business model. In addition, companies are conducting sustainable business practices through the establishment of networks responsible for environmental concerns and the adoption of environmentally friendly innovative technologies. Given the limited resources, the companies invest in and finance sustainable production and environmental protection through measures such as recycling used materials.

Aside from the antecedents, there is a wide range of outcomes that derive from the corporate-level sustainability practices of green entrepreneurship. That is, the practice of SCEE entails dual goals of environmental and business sustainability [30,36]. In general, it can be seen as an engine for economic, social, and environmental sustainability [1,26]

by ensuring balance and harmony between the economy, environment, and society [11]. This is done by preserving natural resources and creating a clean environment [1]. The particular outcomes include innovation [12] and a positive public image [13]. The other macro-level benefits are greener industrial growth [12]; a green economy ([1,31]); reduced negative environmental impacts [1,13,31,38]; social value creation [2]; the development of responsible citizenship [36]; and changing consumer behavior [1,30,38]. In general, the introduction of SCEE leads to a sustainable business in terms of economic return, social responsibility, and a protected environment. The practice also enhances overall improvements in green innovation and industrial growth and the development of a greener economy. The other benefits are a clean environment, minimized negative environmental impact, corporate and public image, and improved consumer behavior.

The outcomes of adopting SCEE at the enterprise level are associated with improved enterprise growth [9]; improved competitive advantage [1,24,26,56]; and the improved handling of toxic materials and a sustainable recycling process [1,41]. It also ensures a sustainable financial system [39]; clean manufacturing processes [15]; better business performance and economic returns ([7,28,31,36,39,40,58]; adaptation to change [10]; and eco-friendly products [13]. In general, the inclusion of ecological aspects in corporate strategy and business decisions will have a positive impact on the sustainability of companies.

### 4.4. The SCEE Practices among the Different Sizes of Organizations

In terms of size, both small and large companies have to take ecological aspects into account when making business decisions. Because of their large volume, SMEs can potentially serve as key drivers of green innovation or innovative practices that reduce environmental damage from business activities ([13]). SMEs are responsible for a significant share of resource consumption, air, water pollution, and waste generation, although large companies, driven by external pressures, focus more on a strategic CSR approach than small companies [23]. In addition, organizations such as NGOs play an increasingly important role in a changing world made fragile by the intense consumption of natural resources that depletes biological reserves [11]. In addition, the outcome of this review suggests that there is greater external pressure on larger companies to take green action. Although small businesses lack the resources to practice green entrepreneurship, the review's findings point to the need for small business involvement in adopting sustainable green entrepreneurship. This is because the cumulative effect of a large number of small companies damages the environment more than the number of large companies. In addition, small businesses are participating in discovering the market opportunities arising from greener business practices, as well as the cost reductions gained from practicing ecological entrepreneurship. Further, entrepreneurs in small firms are more innovative in the efficient use of natural resources and carrying out managerial sustainable practices.

### 5. Conclusions and Implications

This study presents a comprehensive framework for SCEE by incorporating various scattered concepts from different disciplines. The study particularly indicates the major concepts related to the new construct of SCEE, such as green entrepreneurship, green innovation, business environment turbulence, green economy, green business, green product innovation, sustainable entrepreneurship, and sustainable supply chain management. The study also identifies the determinants of SCEE and classifies them under pull and push factors. Further, it indicates the execution processes as well as the firm-level and macro-level consequences of SCEE. Moreover, the study compares the ecological entrepreneurship practices of small and larger firms.

Given the importance of integrating environmental aspects into business decisions, ecological concerns should be incorporated into the corporate goals of organizations. Accordingly, different studies made recommendations for different institutions and practitioners. For example, Piwowar-Sulej et al. [3] proposed exploration factors influencing entrepreneurial action and the importance of different units in the development of sustain-

able entrepreneurship, while Rezaei et al. [50] recommended small- and medium-sized firm participation. In particular, Ye et al. [15] suggested promotion approaches, such as raising awareness, simplifying legal procedures, training the workforce, and monitoring the waste of non-green companies.

Allen and Malin [16] showed innovative models for integrating green business with environmental concerns and natural resource management. These revealed that the various themes included incorporating environmental aspects into corporate strategy, linking business practices with sustainability aspects, encouraging companies of all sizes to engage in green business, and providing the necessary support in the form of legal awareness, and process simplification and the provision of training, should be the main issues to be considered by policy makers at different levels. In addition, the management of companies of all sizes should consider the above concerns when formulating organizational policies and strategies. In addition, there are studies (e.g., [53]) that drive the strengthening of policies that support ecological environmental protection and reduce pollution by inspiring ecological problems, reducing entrepreneurship, and starting environmentally oriented enterprises. In addition to policies, these require the government to work on raising awareness in society and creating incentives for outstanding companies in environmental protection.

Practitioners can use the identified configurations of sustainable business model components as an inspiring starting point for the development of novel sustainable business models [2]. In this context, companies should develop a new business model that links value creation and collection activities with ecological and sustainable aspects. Aside from integrating environmental sustainability into a business model, managers should set examples of pro-environmental behavior to protect the natural environment ([17]). That is, they must be willing to encourage eco-innovation that raises the level of environmentalism to create an adhocracy philosophy and culture [58]. That is, management should work to raise awareness of environmental issues and build an organizational culture that promotes environmental protection.

Regarding the future research agenda, there are limits to researching SCEE. Cri-ado-Gomis et al. [27] noted that most of the previous papers are conceptual and theoretical, requiring more papers with validity and reliability through their empirical verification. Therefore, this study has limitations, since it is difficult to ensure the validity and reliability objectively. This study is also a theoretical one that did not undertake empirical verification. Future researchers should, therefore, expand the subject of study to empirically test and develop measuring instruments for the new construct of SCEE. Empirical validation is required to understand if, how, and to what extent eco-innovation is important for both green-oriented and non-green-oriented SMEs [58]. A more comprehensive, in-depth analysis of individual cases in different economies can also help to uncover the complexities of developing and implementing green strategies [37]. Furthermore, this review only focused on corporate enterprises, while considering practices at national, regional, sectoral, and individual levels is of paramount importance. Although this review focused on corporate organizations to address the issues in detail, considering the multidimensional aspects of corporate ecological entrepreneurship is paramount to emerging as a new research agenda. That said, it would be of great interest to study these types of ecopreneurs in their immediate environments [10]. Therefore, it is of utmost importance to examine the companies and their environment at the same time.

Future research would benefit significantly from integrating different theoretical paradigms such as risk-bearing theory, dynamic capability theory, NRBV, TPB, stakeholder theory, neo-institutional theories, and ecological modernization theory to study ecological and corporate entrepreneurial sustainability. Given these, future researchers should create an integrated framework. Aside from establishing such a comprehensive framework of SCEE, future research should aim to identify the potential similarities and differences between the different schools of thought and definitions described so far in this research field. Therefore, future researchers should work to clarify the concept and definition of SCEE by integrating concepts from different theories and literature.

Golsefid-Alavi [1] identified both the internal and external factors influencing the green entrepreneurial direction of firms and provided recommendations for future re-searchers to use and expand their studies in this field. Accordingly, Tshiaba et al. [24] provide a detailed conclusion for the research to include government policies and demographics as control variables. Although recently, Alwakid et al. [40] recommended that future researchers should test the relationship between green entrepreneurship and sustainable development using different proxies for social, environmental, and economic aspects. Further research should consider specific factors such as norms, culture, trust, and power that could influence both the sustainable innovation climate and economic outcomes [41]. In particular, for the subject-related variables, researchers propose certain variables; for example, demographic variables such as educational level, age, and gender [8], and organizational citizenship behavior on the environment [17]. Given the addition of the contributions of various researchers, future researchers have to consider the role of a variety of aspects related to corporate ecological entrepreneurship, including sustainable innovation, organizational policies, organizational cultures, demographic factors, etc. This is because the previous studies were mainly related to the macro-level aspects, while the contribution of specific company-level variables is still understudied.

The methodology can be improved by indicating the demographic characteristics, including origin (urban or rural) and respondents, as well as including qualitative and quantitative analysis in the survey, as a qualitative survey can enrich the results for future studies. Urbaniec and Żur [42] also added that future studies should use previous evidence to conduct quantitative and theoretical testing research on these and other factors of effectiveness within business creation accelerators. Accordingly, the methodological aspects of investigating CEE sustainability should not be limited to either qualitative or quantitative ones, as both theoretical and empirical studies are still in their infancy. This requires that future researchers have a variety of ways to clarify and work more on CEE sustainability. In addition, Khan et al. [17] proposed longitudinal data to carefully analyze the conclusions and a larger sample size to validate the report's empirical findings. Similarly, future research should study the transformation of sustainable businesses through digital technologies [2], in order to initiate changes in environmental behavior [56].

The study of SCEE is not sufficiently addressed in developing countries, such as countries in Africa. Regarding firm size, Purwandani and Michaud [13] found that using a theoretical framework could help provide foundations or testable concepts in future studies of small business sustainability. Regarding the tourism industry, Luu [51] stated that organizations in the tourism industry should promote the green creativity of their employees to create sustainable tourism services by cultivating a green entrepreneurial direction. This indicates that future researchers should consider different sizes and types of organizations when studying the sustainability of green entrepreneurship. Others in-dicate that examining the role of institutions as facilitators of the relationship between eco-innovation and environmental performance offers a great opportunity for further re-search [11]. In addition, it would be important to review the critical factors that enable effective collaboration between companies and start-ups [42]. Furthermore, future research should consider a broader sample of cases, including organizations from different countries, to examine how different cultural contexts can influence the results [23]. Moreover, future research should examine different economic indicators and larger geographical regions. To conclude, there are multiple opportunities for future researchers to expand the CEE sustainability in terms of its application among corporations and small start-ups, organiza-tional roles, companies' eco-innovation, operational performance, organizational culture in different contexts, and a firm's EO. Further, future research will have to identify the links between individual, organizational, and contextual levels.

**Author Contributions:** Conceptualization, Y.L., A.K.C., S.D. and S.B.N.; methodology, Y.L., A.K.C., S.D. and S.B.N.; formal analysis, Y.L., A.K.C., S.D. and S.B.N.; investigation, Y.L., A.K.C., S.D. and S.B.N.; writing—original draft preparation, Y.L., A.K.C., S.D. and S.B.N.; writing—review and editing,

Y.L., A.K.C., S.D. and S.B.N.; visualization, Y.L., A.K.C., S.D. and S.B.N. All authors have read and agreed to the published version of the manuscript.

**Funding:** This research received no external funding.

**Institutional Review Board Statement:** Not applicable.

**Informed Consent Statement:** Not applicable.

**Data Availability Statement:** The contributions presented in this study are included in the article, further inquiries can be directed to the corresponding author.

**Acknowledgments:** The first author acknowledges Guangzhou Philosophy and Social Science Development Planning Project "Research on Guangzhou's Construction of a Strong City with Advanced Manufacturing Industry from the Perspective of Regional Synergy and Policy Adaptation" (No. 2022GZYB36). The second and the third authors acknowledge University of Johannesburg for covering costs of publication.

**Conflicts of Interest:** The authors declare no conflict of interest.

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
