# Peer review of "The Paradox in the Ecological Side of Corporate Entrepreneurship Sustainability: A Research Agenda and Policy Direction"

_sustainability, doi:10.3390/su142215198_

Round 1

Reviewer 1 Report

The paper is well-structured and able to bring attention to readers. The research questions are well connected and closely related to the topic. The graphical illustration is clearly presented.

Few concerns before moving to publication:

1.     In section 4.3 last sentence, please double check if it is figure 1 or figure 2. In this manuscript, figure 2 is about Years of publications in section 5.1

2.     Incomplete sentence in second paragraph of section 5.1

3.     More examples around the globe could be taken into reference for exploring further

4.     More in-depth analysis could be provided to support your discussion. Try to discuss more specifically in response to your research topics.

Author Response

Dear Reviewer,

First, we want to forward our acknowledgment and special thanks to all reviewers and editor for allowing us to revise the manuscript. The comments and concerns raised are constructive and contribute a lot to improving the entire paper. Therefore, we made all the necessary modifications as per the comments provided. A line-by-line response for each reviewer's comments has been provided below and the modified parts were indicated in blue color in the main manuscript. 

Comment #1: In section 4.3 last sentence, please double check if it is figure 1 or figure 2. In this manuscript, figure 2 is about Years of publications in section 5.1

Author’s Correction #1: Dear reviewer, the comment is well accepted and modified as per the comment. That is, Figure 2 is changed to figure 1.

Comment #2: Incomplete sentence in second paragraph of section 5.1

Author’s Correction #2: Thank you for the comment. Now we have modified the sentence in the main manuscript indicated with a track change.

Comment #3: More examples around the globe could be taken into reference for exploring further

Author’s Correction #3: As per the comment, examples from around the globe is included in page 10.

Comment #4: More in-depth analysis could be provided to support your discussion. Try to discuss more specifically in response to your research topics.

Author’s Correction #4: Dear reviewer, the comment is well accepted and detailed discussion is added under the discussion section. Besides, the previous arguments that will address this concern is indicated with a blue color for clarity. 

Reviewer 2 Report

·       -  Very good paper, and very relevant given the current landscape.  

·      -  This is study provides valuable information and context to this very relevant topic. Its publication can contribute to better and broader understanding of the issues. I recommend this for publication.

·       - While I may have observed some very minor concerns or weaknesses, I do not find that these are important or significant enough to merit a major revision or hinder its publication.

T-The topic, and the the way the research has been laid out, is – in myassessment – foundational to our understanding of the topic and its issues.

·      -  The methods are impressive. The thoroughness imbued in the quality of the literature review, the analysis conducted, and the insights generated are quite rigorous and in-depth.

Author Response

Dear Reviewer,

First, we want to forward our acknowledgment and special thanks to all reviewers and editor for allowing us to revise the manuscript. The comments and concerns raised are constructive and contribute a lot to improving the entire paper. Therefore, we made all the necessary modifications as per the comments provided. A line-by-line response for each reviewer's comments has been provided below and the modified parts were indicated in blue color in the main manuscript. 

Comment #1: Very good paper, and very relevant given the current landscape.  This is study provides valuable information and context to this very relevant topic. Its publication can contribute to better and broader understanding of the issues. I recommend this for publication. While I may have observed some very minor concerns or weaknesses, I do not find that these are important or significant enough to merit a major revision or hinder its publication. T-The topic, and the the way the research has been laid out, is – in my assessment – foundational to our understanding of the topic and its issues. The methods are impressive. The thoroughness imbued in the quality of the literature review, the analysis conducted, and the insights generated are quite rigorous and in-depth

Author’s Correction #1: Dear reviewer, we appreciate for the time taken to review our paper. In order to address the few concerns you raised and comments given by other reviewers, we have tried to modify the paper as indicated in the main manuscript.

Reviewer 3 Report

1.     There is no need to mention the separate heading of contribution, I feel that it should be a part of introduction.

2.     No justifications or economic reasons are mentioned for empirical results. The discussion is written vague. Its hard to follow.

3.     The summary of results should be specific, instead of general. What are the key findings of this study.

4.     Policy recommendations and future research agendas should be in the conclusion section, without any heading.

Author Response

Dear Reviewer,

First, we want to forward our acknowledgment and special thanks to all reviewers and editor for allowing us to revise the manuscript. The comments and concerns raised are constructive and contribute a lot to improving the entire paper. Therefore, we made all the necessary modifications as per the comments provided. A line-by-line response for each reviewer's comments has been provided below and the modified parts were indicated in blue color in the main manuscript.

Comment #1: There is no need to mention the separate heading of contribution, I feel that it should be a part of introduction.

Author’s Correction #1: Dear reviewer, the comment is well accepted and the contribution is merged to the introduction section.

Comment #2: No justifications or economic reasons are mentioned for empirical results. The discussion is written vague. It’s hard to follow.

Author’s Correction #2: As per the comment, we have added reasons for the results and further discussions under section 5 of the manuscript starting from page 10. In addition, we have indicated the arguments that will address this concern in blue color.

Comments #3: The summary of results should be specific, instead of general. What are the key findings of this study?

Author’s Correction #3: the comment is well accepted and the major findings are included in the conclusion section.

Comment #4:  Policy recommendations and future research agendas should be in the conclusion section, without any heading.

Author’s Correction #4: Dear reviewer, modification was made as per the comment, the recommendation and future research agenda sections are merged to the conclusion section.

Reviewer 4 Report

This paper evaluated in the ecological side of corporate entrepreneurship sustainability. This paper's content can inspire further work for researchers dealing with  of ecological and entrepreneurial sustainability in the context of corporate enterprise.

Authors should have some clear recommendations for future works related to this topic.

·  What are the study limitations?

- References should be changed as they are inconsistent (e.g. lowercase letters of surname etc).

Congratulations for the work done.

Author Response

Dear Reviewers,

First, we want to forward our acknowledgment and special thanks to all reviewers and editor for allowing us to revise the manuscript. The comments and concerns raised are constructive and contribute a lot to improving the entire paper. Therefore, we made all the necessary modifications as per the comments provided. A line-by-line response for each reviewer's comments has been provided below and the modified parts were indicated in blue color in the main manuscript. 

This paper evaluated in the ecological side of corporate entrepreneurship sustainability. This paper's content can inspire further work for researchers dealing with of ecological and entrepreneurial sustainability in the context of corporate enterprise.

Comment #1: Authors should have some clear recommendations for future works related to this topic.

Author’s Correction #1: Dear reviewer, the comment is well accepted. The recommendation for the future researchers are clearly indicated with blue color within the last five paragraphs of the conclusion and implication sections.

Comment #2: What are the study limitations?

Author’s Correction #2: as per the comment, the limitations of this study is included and indicated with a blue color in the conclusion and implication section.

Comment #3: References should be changed as they are inconsistent (e.g. lowercase letters of surname etc).